# Longitudinal Predictors of Coronavirus-Related PTSD among Young Adults from Poland, Germany, Slovenia, and Israel

**DOI:** 10.3390/ijerph19127207

**Published:** 2022-06-12

**Authors:** Dominika Ochnik, Aleksandra M. Rogowska, Ana Arzenšek, Joy Benatov

**Affiliations:** 1Faculty of Medicine, University of Technology, 40-555 Katowice, Poland; 2Institute of Psychology, University of Opole, 45-052 Opole, Poland; arogowska@uni.opole.pl; 3Faculty of Management, University of Primorska, 6101 Koper, Slovenia; ana.arzensek@fm-kp.si; 4Department of Special Education, University of Haifa, Haifa 3498838, Israel; jbentov2@gmail.com

**Keywords:** mental health, COVID-19, coronavirus-related PTSD, young adults, trust in institutions

## Abstract

The aim of this study was to reveal longitudinal predictors of coronavirus-related PTSD and the moderating roles of country, sex, age, and student status among young adults from Poland, Germany, Slovenia, and Israel. We included the following predictors: perceived stress, exposure to COVID-19, perceived impact of COVID-19 on well-being in socioeconomic status (PNIC-SES) and social relationships (PNIC-SR), fear of COVID-19, fear of vaccination, and trust in institutions. We conducted the study online among a representative sample of 1723 young adults aged 20–40 (*M* = 30.74, *SD* = 5.74) years in February 2021 (T1) and May–June 2021 (T2). We used McNemar’s *χ*^2^ and the paired samples Student’s *t*-test to test differences over time. We assessed the relationships between variables using Pearson’s correlation. We performed structural equation modeling (SEM) to examine the associations between variables at T1 and T2. We used a lagged regression model to examine the causal influences between variables across different time points (T1 and T2). The results showed that all variables decreased over time, except exposure to COVID-19. The rates of infected, tested, and under-quarantine participants increased. The rates of those who lost a job and experienced worsening economic status decreased. The rate of hospitalized participants and those experiencing the loss of close ones did not change. Higher perceived stress, fear of COVID-19, fear of vaccination, and trust in institutions were significant longitudinal predictors of coronavirus-related PTSD regardless of country, sex, age, and student status. Institutions should provide more accurate programs for public health, so trust in institutions can be a protective and not a risk factor in future traumatic events.

## 1. Introduction

The unexpected rapid global spread of coronavirus disease 2019 (COVID-19) has negatively affected mental health [1,2,3,4,5,6,7]. Epidemiological research has shown a psychiatric epidemic co-occurring with the COVID-19 pandemic [8].

The groups most vulnerable to mental health deterioration in the general population are young adults [5,9,10,11,12], particularly students [13,14,15] and women [15,16,17,18,19,20]. The cross-cultural context is also influential in the prevalence of mental health issues [14,15]. Furthermore, COVID-19 survivors have experienced mental health problems more profoundly than those not infected [21,22]. However, even those not exposed to COVID-19 infection have directly experienced mental health deterioration [23].

Post-traumatic stress disorder (PTSD) refers to the development of specific negative symptoms due to exposure to a traumatic event. Symptoms may involve fear-based re-experiencing, behavioral and emotional changes, adverse effects on cognition, and dysphoric moods [24]. A discussion is ongoing as to whether exposure to COVID-19 can be treated as a traumatic experience and meet the PTSD criteria in the DSM-5 [25]. However, experiences with the COVID-19 pandemic have been recognized as a new type of traumatic stressor that has severe mental health effects, including PTSD [26,27]. The overall prevalence of PTSD during the pandemic was 17.68% [28]. The prevalence of coronavirus-related PTSD risk at the three cutoff scores of 25, 44, and 50 was 78.20%, 32.70%, and 23.10%, respectively, among young adults from six countries [25].

Female sex, lower perceived economic stability, and fear of contagion of COVID-19 have been determined as risk factors of PTSD [29]. Furthermore, fear of COVID-19 directly affects PTSD via the moderating role of nomophobia and the mediating role of psychological stress [30]. PTSD was more common in those who are younger, female, unmarried, less educated and unemployed during the pre-pandemic period [31], and less educated during the pandemic [10]. A recent cross-national study among university students showed that female sex, a prior diagnosis of depression, a loss of friends and/or relatives, job loss, and worsening economic status due to COVID-19 were positively associated with high and very high coronavirus-related PTSD risk [25]. In contrast, female sex, prior PTSD diagnosis, experiencing COVID-19 symptoms, testing for COVID-19, having infected friends and/or relatives, and worsening economic status were associated with moderate risk [25]. However, female sex, subjective socioeconomic status, and the number of COVID-19 cases were not found to be PTSD risk factors in a sample of Chinese university students [32,33]. Furthermore, a recent meta-analysis showed that sex was not a risk factor for COVID-19 during the pandemic [34]. Therefore, the role of sex in PTSD related to the COVID-19 pandemic is unclear.

PTSD is related to perceived stress; however, perceived stress and PTSD symptoms have unique effects on cognitive functioning [35,36].

Diagnosis of PTSD and vaccine hesitancy levels are positively related in clinical practice [37], in contrast to clinical depression and anxiety levels [38]. However, the concept of fear of vaccination [39,40] has rarely been explored in comparison to vaccination hesitancy [41,42,43]. Fear of vaccines is strongly predicted by political ideology [44]. Additionally, trust in institutions is a positive predictor of fear of COVID-19 [45]. Institutions failed to comfort their citizens during the COVID-19 pandemic, and those with increased trust in institutions experienced elevated fear of COVID-19 [45]. Trust in institutions is a part of the social capital concept [46], and social capital is indirectly related to PTSD [47]. Low levels of social capital are related to increased levels of anxiety and stress [48]. However, little is known about the relationships between trust in institutions and coronavirus-related PTSD. We aimed to fill the research gap on the relationships between trust in institutions and fear of vaccination with coronavirus-related PTSD. The results can be used to build better public health programs.

The aim of this study was to reveal the role of exposure to COVID-19 and several mental health indices, namely, perceived stress, exposure to COVID-19, perceived negative impact of COVID-19 on socioeconomic and social relationship aspects of well-being, fear of COVID-19, fear of vaccination, and trust in institutions as longitudinal predictors, in coronavirus-related PTSD in a representative sample of young adults in a cross-national context. Poland, Germany, Slovenia, and Israel represent a variety of global cultural values [49]. Moreover, we aimed to reveal the moderating role of country, sex, age, and student status in the proposed longitudinal model of coronavirus-related PTSD. We also aimed to evaluate changes in exposure to COVID-19 over time. We expected the variables would be longitudinal predictors in a three-month period on the verge of the third wave of the pandemic.

## 2. Materials and Methods

### 2.1. Study Design

We conducted this study with a longitudinal design among representative samples of young adults (20–40 years old) from Poland, Germany, Slovenia, and Israel at two measurement time points. The first measurement (T1) was between 19 and 26 February 2021, and the second measurement (T2) was between 26 May and 9 June 2021. The survey was administered online by the ARIADNA panel. The inclusion criteria were age (20–40 years) and country (Poland, Germany, Slovenia, or Israel). The participants answered the questions in their native language. Translators with experts from four countries translated the survey questions from English based on the cross-cultural adaptation guidelines [50].

Acquiescence biases were controlled by providing a variety of possible responses, both visually and regarding the content of answers. The survey modified the answers from higher to lower and from lower to higher to void biases. The participants could stop at any moment and return to finish the survey when desired. We placed no time limit on survey completion. The average time to complete the online survey was 21.52 min (SD = 136.75). The total number of participants in T1 was 2951; during T2, the number was 1724. Due to anomaly detection, one observation was excluded from T2. Overall, the response rate was 58.42% in T2. Therefore, the final sample comprised 1723 respondents who participated in both time points of measurement.

The pandemic during this three-month period substantially improved. In all countries, the number of new COVID-19 cases and deaths due to COVID-19 decreased as the number of vaccinated people increased. Furthermore, the stringency of restrictions dropped [40,45].

This study was a part of the international project “Mental health of young adults during the COVID-19 pandemic in Poland, Germany, Slovenia, and Israel: A longitudinal study” [51].

### 2.2. Participants

The sample consisted of 1723 adults from Poland (*n* = 446, 26%), Germany (*n* = 418, 24%), Slovenia (*n* = 431, 25%), and Israel (*n* = 428, 25%). Just over half of the total sample constituted women (54%, *n* = 935), 75% lived in towns or cities (vs. villages) (*n* = 1297), 58% were childfree (*n* = 1001), 29% coupled (*n* = 505), and 71% were employed (*n* = 1218). A current student status was reported by 24% of the total sample (*n* = 420). The mean age of the participants was 31 (*M* = 30.74, *SD* = 5.74) years. A detailed description of the demographic characteristics of the study sample is presented Appendix A.

### 2.3. Measurements

#### 2.3.1. Coronavirus-Related PTSD

The coronavirus-related PTSD was measured by the 17-item PTSD Checklist-Specific Version (PCL-S) [52]. Considering that we referred to a particular event of the lockdown during the COVID-19 pandemic, PCL-S was the best-fitted instrument. To be highly precise, the expression “a stressful experience from the COVID-19 lockdown” was added to each of the 17 items. The scale utilizes a Likert scale ranging from 1 to 5, where 1 = not at all; 2 = a little bit; 3 = moderately; 4 = quite a bit; 5 = extremely. The total sum is between 17 to 85 scores. A higher total score means a higher coronavirus-related PTSD. The test score reliability coefficient showed high reliability, with Cronbach’s α 0.96 at T1 and T2.

#### 2.3.2. Perceived Stress

We employed the Perceived Stress Scale (PSS-10) [53] to measure whether the respondents appraised the situation in their life as stressful. The PPS-10 consists of 10 items related to the frequency of stressful events in the month preceding the study on a five-point scale (0 = never to 4 = very often). Cronbach’s α for this sample was 0.83 in T1 and T2.

#### 2.3.3. Perceived Negative Impact of the COVID-19 Pandemic

The Perceived Negative Impact of Coronavirus (PNIC) on well-being evaluates the main sources of concerns during the COVID-19 pandemic [25]. Participants rated statements on a five-item Likert scale (from 1 = strongly disagree to 5 = strongly agree) to what extent the current situation associated with the COVID-19 pandemic may negatively affect their life. The first subscale refers to socioeconomic status (PNIC-SES) in the following aspects: (1) Completing the semester and graduation, (2) finding a job and professional development, and (3) financial situation. The second subscale refers to social relationships (PNIC-SR): (1) With loved ones and family, and (2) with colleagues and friends. The higher the scores, the more serious the coronavirus-related concerns were perceived to be in the given aspect of life. The Cronbach’s α coefficient for PNIC-SES was 0.81 in T1 and 0.86 in T2, while for PNIC-SR it was 0.84 in T1 and 0.88 in T2.

#### 2.3.4. Fear of COVID-19

The Fear of COVID-19 Scale (FCV-19S) is a seven-item scale for evaluating fear of COVID-19 [54]. The participants answer on a Likert-type scale, ranging from 1 = strongly disagree to 5 = strongly agree. The total result varies between 7 and 35. A higher score reflects greater fear of COVID-19. Cronbach’s α for this scale was 0.91 at T1 and 0.92 at T2 in the study.

#### 2.3.5. Fear of Vaccination

The seven-item Fear of Vaccination Scale (FoVac) [40,51] is based on FCV-19S [55]. We changed the term “Corona” to “vaccination for COVID-19” for all items. FoVac uses a Likert-type scale (from 1 = strongly disagree to 5 = strongly agree), analogically to FCV-19S. The sum of scores ranges from 7 to 35. A higher score relates to greater fear of vaccination for COVID-19. Cronbach’s α for fear of vaccination was 0.92 at T1 and 0.93 at T2 in the study.

#### 2.3.6. Trust in Institutions

The Trust in Institution Scale is a three-item scale related to trust in parliament, trust in the legal system, and trust in politicians. It forms a part of the social capital of The European Social Survey [53]. The participants answer on an 11-point scale, ranging from 0 = no trust at all to 10 = complete trust. A higher score means a higher trust in institutions. Cronbach’s α for this scale in this study was 0.88 at T1 and 0.87 at T2.

#### 2.3.7. Self-Reported Exposure to COVID-19

The exposure to COVID-19 [25,51] was based on eight items related to the COVID-19 pandemic: (1) symptoms that may indicate coronavirus infection; (2) being tested for COVID-19; (3) hospitalization due to COVID-19; (4) experiencing strict quarantine for at least 14 days, in isolation from loved ones due to COVID-19; (5) coronavirus infection among family, friends, or relatives; (6) death among relatives due to COVID-19; (7) losing a job due to the COVID-19 pandemic—the person or their family; and (8) experiencing a worsening of economic status due to the COVID-19 pandemic. The participants answered on a dichotomous scale, where 0 = no and 1 = yes. Each aspect of the exposure to COVID-19 was separately analyzed as a dichotomous variable in analyses of change and as a continuous variable (sum of points) in the change model and model of prediction for coronavirus-related PTSD. The self-exposure to COVID-19 items were developed based on the methodology proposed by Tang et al. [33].

#### 2.3.8. Sociodemographic Data

The sociodemographic data included sex, age (20–30 or 31–40 years), place of residence (village or town/city), employment status (employed or unemployed), relationship status (single or coupled), and having children (with children or childfree).

### 2.4. Statistical Analyses

We calculated changes over time in particular items of exposure to coronavirus using McNemar’s *χ*^2^. We constructed a series of 2 × 2 contingency tables to examine the marginal homogeneity that the proportions of adults exposed to coronavirus changed (increased or decreased) between T1 and T2. McNemar’s *χ*^2^ test provides population-averaged estimates of change over time for binary data, testing the hypothesis that the average disparity between two time points is different from zero [56]. We calculated the odds ratio (OR) (quotient B/C) as an effect size, with an interpretation of 1.22 as small, 1.86 as medium, and 3.00 as large effect sizes [57].

The initial analysis of descriptive statistics showed good psychometric properties for PTSD, perceived stress, exposure to COVID-19 (as a continuous variable as a sum of all eight items), perceived negative impact of coronavirus (PNIC-SES and PNIC-SR), fear of COVID-19, fear of vaccination, and trust in institutions at T1 and T2, which were required for parametric tests. Therefore, we conducted parametric analyses in the next steps of the study. We used the paired samples Student’s *t*-test to test differences in all continuous variables (coronavirus-related PTSD, stress, exposure, PNIC-SES, PNIC-SR, fear of COVID-19, fear of vaccination, and trust in institutions) between T1 and T2. We used Cohen’s *d* coefficient to assess effect size (small for *d* = 0.20, medium for *d* = 0.50, and large for *d* = 0.80).

We assessed the relationships between variables using Pearson’s correlation. Additionally, we used the lagged regression model to examine the causal influences between variables across different time points (T1 and T2) during the COVID-19 pandemic. We examined the stability and relationships between variables over time to better understand how all variables at T1 (including coronavirus-related PTSD, perceived stress, exposure to COVID-19, PNIC-SES, PNIC-SR, fear of COVID-19, fear of vaccination, and trust in institutions) influenced PTSD at T2. In addition to allowing for the estimation of regression effects, lagged path models also controlled for associations within time points and autoregressive effects, or stability, across time. Autoregressive effects describe the amount of stability in constructs over time. We evaluated all structural models using several goodness-of-fit criteria [58], including maximum likelihood (ML) *χ*^2^, *df*, *p*-value (a χ^2^/*df* ratio of <5 representing good fit), standardized root mean squared residual (SRMR < 0.08 is acceptable), root mean square error of approximation (adequate fit if RMSEA ≤ 0.08), and comparative fit index (CFI ≥ 0.90 meaning adequate fit).

We examined the measurement invariance (MI) by multigroup comparison (MGMI) to check whether the lagged regression model varied across countries (Germany, Israel, Poland, and Slovenia), sexes (women and men), age groups (younger adults between 20 and 30 years and older adults between 30 and 40 years), and student status (student and non-student). The MGMI analysis assumes hierarchical tests for invariance of measurement parameters, with more equality restrictions in configural and metric models. For an adequate sample size (*n* > 300), Chen [59] suggested a change of >−0.010 in CFI, supplemented by a change of >0.015 in RMSEA or a change of >0.030 in SRMR to indicate non-invariance. For testing intercept or residual invariance, a substantial change that suggests non-invariance is >−0.010 in CFI, supplemented by a change of >0.015 in RMSEA or a change of >−0.010 in SRMR. We performed McNemar’s *χ*^2^ test using STATISTICA ver. 13.3 [60]. The other statistical analyses, as well as Figure 1, were performed and drawn using JASP Team [61], except SEM analysis with MGMI (Figure 2) conducted by AMOS ver. 23.

We used G*Power [62] to calculate an appropriate sample size. For McNemar’s *χ*^2^ test, using the odds ratio and the proportion expected to change, a two-tailed, 1.5 effect size at an alpha of 0.05, and 95% power and proportion discordant pairs of 0.30, we determined that a minimum of 148 participants would be needed to detect an effect. For the matched pairs Student’s *t*-test, two-tailed, with Cohen’s *d* = 0.50 effect size, at an alpha of 0.05 and 95% power, a minimum of 54 participants was necessary. For repeated measures two-way mixed factor analysis of variance (ANOVA) the expected sample size was 158, assuming the two groups and two measurements, effect size *f*^2^ = 0.25, repeated measures *r* = 0.50, *p* < 0.05, and 95% CI. For bivariate correlation (two tails), the necessary sample size was 138 people, if *r* = 0.30, *p* < 0.05, and 95% CI. When we considered the linear multiple regression model for two independent variables, with an effect size *f*^2^ = 0.15, *p* < 0.05, and 95% CI, the expected sample size was 107 people. For structural equation modeling (SEM) analysis with multigroup measurement invariance (MGMI), the adequate total sample had to exceed 300 people [59].

#### Common Method Bias

We implemented Harman’s single factor test to identify whether the common method bias significantly affected the study variables. All 57 items from all measurement tools (i.e., PCL-17, PSS-10, Exposure-8, PNIC-5, FoC-7, FoV-7, and Trust-3) were included in the exploratory factor analysis (EFA) for one fixed factor, with maximum likelihood (ML) extraction method, and no rotation. The Bartlett’s test of sphericity was χ^2^(1596) = 66,925, *p* < 0.001; while KMO measure of sampling adequacy was 0.959 (KMO > 0.8), both indicating that there is substantial correlation in the data and factor analysis is useful to detect the multifactor structure. The single factor explained 28.2% of variance, and fit indices for the one factor model were unacceptable: χ^2^(1539) = 34746, χ^2^/*df* = 22,577, *p* < 0.001, RMSEA = 0.112 (95% CI = 0.111, 0.113), TLI = 0.473, BIC = 23,278. In comparison, parallel EFA with ML extraction method and varimax rotation showed 11 factor solution, which explains 60% of variance. Fit indices suggest acceptable adjusting to the model: χ^2^(3947) = 1024, χ^2^/*df* = 3.85, *p* < 0.001, RMSEA = 0.041 (95% CI = 0.039, 0.042), TLI = 0.930, BIC = –3683. Since multifactor EFA solution is better that single factor model, we can conclude that the common bias is not problem in the study.

## 3. Results

### 3.1. Changes in Variables over Time

#### 3.1.1. Changes in Coronavirus-Related PTSD and Its Predictors over Time

We performed a paired samples Student’s *t*-test to examine changes over time in mental health indices. The results are shown in Table 1. We found a significant decrease at T2 in PTSD, perceived stress, perceived negative impact of coronavirus (PNIC) in both the socioeconomic status (SES) and social relationships (SR) dimensions, fear of COVID-19, fear of vaccination, and trust in institutions (as a single dimension of social capital) compared to T1. The effect size was small for most of the variables, except for the large effect sizes for fear of COVID-19 and fear of vaccination. Only exposure to the COVID-19 pandemic increased during the second measurement, but these differences were insignificant. The detailed statistics are presented in Table 1.

#### 3.1.2. Changes in Exposure to COVID-19 over Time

Exposure to coronavirus was higher (the rate of infected people) at T2 (28%) than at T1 (26%) (McNemar’s *χ*^2^(1) = 4.73, *p* = 0.030, OR = 1.29). A significantly higher rate of respondents had tested for coronavirus by T2 (36% at T1 and 48% at T2; McNemar’s *χ*^2^ = 94.05, *p* < 0.001, OR = 2.65), and had isolated in strict quarantine for at least 14 days due to COVID-19 (19% at T1 and 21% at T2; McNemar’s *χ*^2^ = 10.53, *p* = 0.001, OR = 1.52). Furthermore, a slightly higher rate of participants indicated experiencing COVID-19 among their friends or relatives at T2 (53%) than at T1 (50%) (McNemar’s *χ*^2^ = 9.43, *p* = 0.002, OR = 1.35).

In contrast, a significantly lower rate of participants or their loved ones had lost their job due to the COVID-19 pandemic by T2 (25% at T1 and 22% at T2; McNemar’s *χ*^2^ = 13.80, *p* < 0.001, OR = 0.65). Moreover, the rate of respondents experiencing worsening of their economic status due to the pandemic significantly decreased from 42% at T1 to 36% at T2 (McNemar’s *χ*^2^ = 32.82, *p* = 0.001, OR = 0.56). The rate of adults hospitalized for COVID-19 also decreased by T2 (3%) compared to at T1 (2%), but these differences were insignificant (McNemar’s *χ*^2^ = 3.37, *p* = 0.067, OR = 0.65). The rate of participants who lost their family members or friends due to COVID-19 was similar at T1 (10%) and T2 (11%) (McNemar’s *χ*^2^ = 0.36, *p* = 0.547, OR = 1.10).

### 3.2. Change Model and Predictors of Coronavirus-Related PTSD

#### 3.2.1. Correlations between Variables at T1 and T2

We examined the associations between mental health indices, such as coronavirus-related PTSD, perceived stress, exposure to COVID-19, both scales of the perceived negative impact of COVID-19 (PNIC) (socioeconomic status (PNIC-SES) and social relationships (PNIC-SR)), fear of COVID-19, fear of vaccination, and trust in institutions. Figure 1 shows the Pearson’s *r* correlations as a heatmap.

**Figure 1 ijerph-19-07207-f001:**
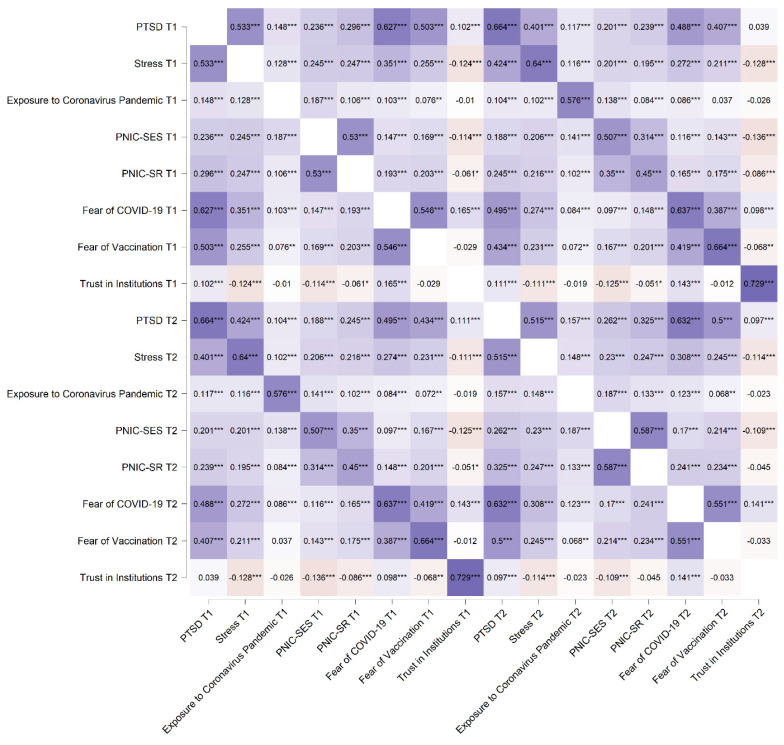
Pearson’s *r* heatmap. *N* = 1723; T1 = time 1 of measurement; T2 = time 2 of measurement; PNIC-SES = perceived negative impact of coronavirus–socioeconomic status; PNIC-SR = perceived negative impact of coronavirus–social relationships. Colors in shades of purple represent positive correlations, while colors in beige negative correlations. The most intense shades of colors represent a large effect size, while the lightest shades—a small effect size. * *p* < 0.05, ** *p* < 0.01, and *** *p* < 0.001.

Almost all of the variables were correlated with weak-to-moderate strength. Coronavirus-related PTSD at T1 was strongly correlated to stress, fear of COVID-19, fear of vaccination at T1 and coronavirus-related PTSD at T2. On the other hand, PTSD at T2 was additionally strongly correlated to stress, fear of COVID-19, fear of vaccination at T2.

Most other variables were positively correlated. In contrast, higher trust in institutions was associated with lower stress, exposure to COVID-19, PNIC-SES, PNIC-SR, and fear of vaccination. However, higher trust in institutions was also related to higher coronavirus-related PTSD and higher fear of COVID-19.

#### 3.2.2. Change Model and Predictors of PTSD at T2

We conducted SEM to examine associations between variables at T1 and T2. The conceptual lagged regression model is presented in Figure 2.

**Figure 2 ijerph-19-07207-f002:**
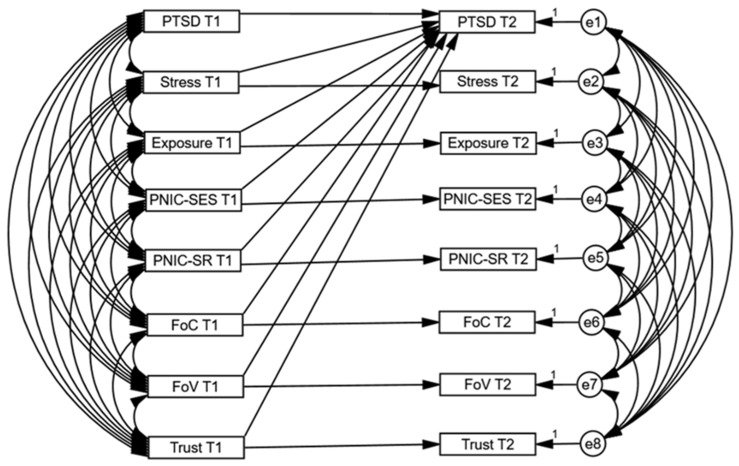
Conceptual model of associations between the variables at T1 and T2 in the lagged regression model (*N* = 1723).

The autoregressive associations of all variables in the model, including coronavirus-related PTSD, perceived stress, exposure to COVID-19, PNIC-SES, PNIC-SR, fear of COVID-19, fear of vaccination, and trust in institutions, were significant, substantial, and positive (Table 2).

A strong autoregressive association indicates slight variance over time, meaning more stability or influence from T1. We found that perceived stress, fear of COVID-19, fear of vaccination, and trust in institutions at T1 were significant positive predictors of coronavirus-related PTSD at T2. However, exposure to COVID-19 and the perceived negative impact of coronavirus (on both scales PNIC-SES and PNIC-SR) were unrelated to coronavirus-related PTSD.

#### 3.2.3. Moderating Role of Country, Sex, Age, and Student Status in the Lagged Regression Model of Coronavirus-Related PTSD at T2 and Mental Health Indices

We separately performed a series of measurement invariance analyses for a country (German, Israel, Poland, or Slovenia), sex (women or men), age (younger adults between 20 and 30 years old or older adults between 30 and 40 years old), and student status (student or non-student) as moderators of the model of the association between coronavirus-related PTSD at T2 and other variables at T1 (Figure 2). The overall model showed an acceptable fit for the *χ*^2^/*df* statistic and an excellent fit considering SRMR, RMSEA, and CFI (baseline model 0 in Table 3).The results of the multigroup measurement invariance (MGMI) are shown in Table 3.

Assuming country, sex, age, and student status as moderators, the results of the MGMI analysis indicated that the same regression model structure (configural invariance) and invariant regression loading (metric invariance). A minimum of two fit indices is recommended to support or reject the invariance hypothesis [59]. Therefore, we concluded that the coronavirus-related PTSD lagged regression model was similarly measured across the country, sex, age, and student status groups. The MGMI was fully confirmed for country, sex, age, and student status because differences between configural and metric models were insignificant [59].

The results showed that country, sex, age, and student status did not play a moderating role in the lagged regression model.

## 4. Discussion

In a three-month period on the verge of the third wave of the COVID-19 pandemic, all of the analyzed variables significantly decreased, except exposure to COVID-19. Our study findings revealed longitudinal predictors for coronavirus-related PTSD in a representative sample of young adults aged 20–40 years from Poland, Germany, Slovenia, and Israel. We found that perceived stress, fear of COVID-19, fear of vaccination, and trust in institutions were significant positive predictors in the coronavirus-related PTSD longitudinal prediction model. However, exposure to COVID-19 and the perceived negative impact of COVID-19 on well-being were insignificant in this model. Moreover, we revealed that country, sex, age, and student status were not moderators of coronavirus-related PTSD in the lagged regression model.

During the study timeline, the pandemic situation considerably improved. The number of deaths and new cases of COVID-19 dropped as many restrictions were relaxed. Furthermore, the number of vaccinated people increased [40,45,55]. Therefore, coronavirus-related PTSD, along with perceived stress and the perceived negative impact of COVID-19 on well-being, significantly dropped over time. We noted the most profound changes in the fear of COVID-19 and fear of vaccination. The young adults across the four countries adapted to the pandemic situation. This is consistent with other findings showing relative stability of perceived stress over time, with more considerable fluctuation in fear of COVID-19 due to change in the COVID-19 mortality rate [63].

Furthermore, during the second measurement, when the number of vaccinated people increased, the fear of vaccination dropped. However, trust in institutions did not increase, but significantly dropping over time instead. Even though previous cross-national studies have shown that trust in institutions increased as the mortality rate decreased [64], our findings did not confirm this.

In our previous analyses, we found that the prevalence of coronavirus-related PTSD significantly dropped from 65% at T1 to 60% at T2 [55]. As we conducted our study over a year after the start of the COVID-19 pandemic, the PTSD rates can be evaluated as long-term consequences of the pandemic. Our findings are in line with those of other studies claiming the COVID-19 pandemic is a traumatic stressor [26,27].

Our findings showed that exposure to the COVID-19 pandemic has increased in terms of new cases, testing, and 14-day quarantine. Job losses due to the pandemic and worsening economic status have significantly decreased. However, we found no changes in the hospitalization or deaths of loved ones due to COVID-19 among the participants. Therefore, even though the number of new cases increased over time, the course of COVID-19 was less severe. Moreover, the exposure to adverse economic consequences of the pandemic lessened.

The main aim of this study was to reveal longitudinal predictors of coronavirus-related PTSD over a three-month period. We showed the predictive role of perceived stress, fear of COVID-19, fear of vaccination, and trust in institutions. We expected that perceived stress would be related to coronavirus-related PTSD [35,36]. However, the roles of fear of COVID-19 and vaccination for coronavirus-related PTSD were unclear. We showed that higher fear of COVID-19 and vaccination predict higher coronavirus-related PTSD. The uncertainty, complexity, and novelty of the COVID-19 pandemic took a toll on people in the form of fear of COVID-19 and vaccination, which affected coronavirus-related PTSD.

Furthermore, those with higher trust in institutions experienced increased coronavirus-related PTSD. This is a clear message that governmental institutions did not sufficiently support their citizens in Poland, Germany, Slovenia, and Israel. Even though previous studies have shown that social capital negatively and indirectly affects PTSD [47], our findings showed that it directly predicts an increase in coronavirus-related PTSD.

Moreover, country, sex, age, and student status did not play a moderating role in the prediction model of coronavirus-related PTSD. Previous findings have shown that these sociodemographic characteristics differentiate mental health issues related to the COVID-19 pandemic [14,15,25,40,45]. However, we revealed that in the longitudinal prediction model of coronavirus-related PTSD, these variables do not moderate the relationships between the predictors and the outcome variable. Therefore, higher perceived stress, fear of COVID-19, fear of vaccination, and trust in institutions positively predict coronavirus-related PTSD regardless of country, sex, age, and student status.

The strength of this study is its longitudinal study design among a representative sample across four countries with diverse values. The novel aspect of this study is revealing the predictive role of fear of vaccination and trust in institutions in coronavirus-related PTSD. The adverse role of trust in institutions in coronavirus-related PTSD should be noticed by institutions across countries. Furthermore, our findings showed that longitudinal predictors affect coronavirus-related PTSD regardless of country, age, sex, or student status.

However, all research measures were based on self-assessment. Therefore, our data may be subject to retrospective response bias. Furthermore, we conducted this study in a specific pandemic situation. Hence, the results might not be generalizable to a non-pandemic situation. Furthermore, we conducted research only at two time points, so conducting a study with several time points to assess the trend would be beneficial.

## 5. Conclusions

Coronavirus-related PTSD significantly decreased over time. Stress, fear of COVID-19, fear of vaccination, and trust in institutions were the risk factors predicting an increase in coronavirus-related PTSD. Therefore, institutions should particularly enhance their work on creating programs for public health, so trust in institutions may be protective and not a risk factor in future traumatic events.

## Figures and Tables

**Table 1 ijerph-19-07207-t001:** Differences in mental health indices between T1 and T2 (*N* = 1723).

Variable	Time 1		Time 2		*t* (1722)	*p*	Cohen’s *d*
	*M*	*SD*	*M*	*SD*			
Coronavirus-related PTSD	39.13	15.93	37.43	15.88	5.42	<0.001	0.13
Perceived stress	19.67	6.59	17.75	5.83	14.96	<0.001	0.36
Exposure to COVID-19	2.12	1.69	2.22	1.67	−2.63	1.000	−0.06
PNIC-SES	3.49	1.07	3.24	1.14	9.29	<0.001	0.22
PNIC-SR	3.28	1.11	3.15	1.18	4.59	<0.001	0.11
Fear of COVID-19	16.09	6.54	8.12	6.53	59.35	<0.001	1.43
Fear of vaccination	16.79	7.11	8.98	7.20	55.32	<0.001	1.33
Trust in institutions	11.02	7.65	10.5	7.50	3.90	<0.001	0.09

Time 1 = time 1 of measurement; Time 2 = time 2 of measurement; PNIC-SES = perceived negative impact of COVID-19–socioeconomic status; PNIC-SR = perceived negative impact of COVID-19–social relationships.

**Table 2 ijerph-19-07207-t002:** Regression of the associations between variables at T1 and T2 (*N* = 1723).

Antecedent at T1	Consequence at T2	*b*	*SE b*	β	*p*
Coronavirus-related PTSD	Coronavirus-related PTSD	0.41	0.02	0.44	<0.001
Perceived stress	Perceived stress	0.54	0.02	0.62	<0.001
Exposure to COVID-19	Exposure to coronavirus	0.56	0.02	0.57	<0.001
PNIC-SES	PNIC-SES	0.47	0.02	0.46	<0.001
PNIC-SR	PNIC-SR	0.41	0.02	0.39	<0.001
Fear of COVID-19	Fear of COVID-19	0.60	0.02	0.62	<0.001
Fear of vaccination	Fear of vaccination	0.62	0.02	0.63	<0.001
Trust institutions	Trust institutions	0.71	0.02	0.73	<0.001
Perceived stress	Coronavirus-related PTSD	0.27	0.04	0.12	<0.001
Exposure to coronavirus	Coronavirus-related PTSD	−0.05	0.14	-0.01	0.699
PNIC-SES	Coronavirus-related PTSD	−0.04	0.26	0.00	0.864
PNIC-SR	Coronavirus-related PTSD	0.28	0.25	0.02	0.255
Fear of COVID-19	Coronavirus-related PTSD	0.25	0.05	0.11	<0.001
Fear of vaccination	Coronavirus-related PTSD	0.17	0.04	0.08	<0.001
Trust institutions	Coronavirus-related PTSD	0.14	0.03	0.07	<0.001

**Table 3 ijerph-19-07207-t003:** Multigroup measurement invariance analysis for country, sex, age, and student status.

Model		ML *χ*^2^	*df*	ML *χ*^2^/*df*	*p*	SRMR	RMSEA	95% CI	CFI	ΔSRMR	ΔRMSEA	ΔCFI
0	Baseline	221.762	49	4.526	<0.001	0.048	0.045	0.039–0.051	0.986			
Country invariance												
1a	Germany	73.758	49	1.505	0.013	0.050	0.035	0.017–0.050	0.990	-	-	-
1b	Israel	122.363	49	2.497	<0.001	0.066	0.059	0.046–0.072	0.974	-	-	-
1c	Poland	94.849	49	1.936	<0.001	0.049	0.046	0.032–0.060	0.985	-	-	-
1d	Slovenia	84.997	49	1.735	<0.001	0.046	0.041	0.026–0.056	0.989	-	-	-
2	MI configural	375.965	196	1.918	<0.001	0.047	0.023	0.020–0.027	0.985	-	-	-
3	MI metric	475.601	241	1.973	<0.001	0.047	0.024	0.021–0.027	0.980	0.000	0.001	–0.005
Gender invariance												
4a	Women	133.535	49	2.725	<0.001	0.043	0.043	0.034–0.052	0.986	-	-	-
4b	Men	129.619	49	2.645	<0.001	0.051	0.046	0.036–0.056	0.986	-	-	-
5	MI configural	263.156	98	2.685	<0.001	0.051	0.031	0.027–0.036	0.986	-	-	-
6	MI metric	285.927	113	2.53	<0.001	0.050	0.030	0.026–0.034	0.985	−0.001	−0.001	−0.001
Age invariance												
7a	Younger	122.422	49	2.498	<0.001	0.048	0.042	0.033–0.052	0.986	-	-	-
7b	Older	145.863	49	2.977	<0.001	0.052	0.047	0.039–0.056	0.986	-	-	-
8	MI configural	268.284	98	2.738	<0.001	0.045	0.032	0.027–0.036	0.986	-	-	-
9	MI metric	316.117	113	2.797	<0.001	0.044	0.032	0.028–0.037	0.983	–0.001	0.000	–0.003
Student status invariance												
10a	Student	104.365	49	2.13	<0.001	0.056	0.052	0.038–0.066	0.982	-	-	-
10b	No student	189.024	49	3.858	<0.001	0.051	0.047	0.040–0.054	0.984	-	-	-
11	MI configural	293.442	98	2.994	<0.001	0.052	0.034	0.030–0.039	0.984	-	-	-
12	MI metric	324.343	113	2.87	<0.001	0.053	0.033	0.029–0.037	0.982	0.001	−0.001	−0.002

ML = maximum likelihood, SRMR = standardized root mean square residual, RMSEA = root mean square error of approximation, CFI = comparative fit index.

## Data Availability

This study formed part of the international research project “Mental health of young adults during the COVID-19 pandemic in Poland, Germany, Slovenia, and Israel: A longitudinal study” [51], registered at the Center for Open Science (OSF). The datasets used and analyzed during the current study are available from the corresponding author upon reasonable request.

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
