# Peer review of "Longitudinal Predictors of Coronavirus-Related PTSD among Young Adults from Poland, Germany, Slovenia, and Israel"

_ijerph, 2022, doi:10.3390/ijerph19127207_

Round 1

Reviewer 1 Report

Thanks for the Editor for giving me this opportunity to review this article. I think this is an important article on Coronavirus-related PTSD. The literature review is very detailed, and the basis for raising issues is very sufficient. The methods are also adequately described. The results are clearly and briefly presented. The discussion was structured, and the conclusions are supported by the results.

I only have a few minor comments.

1)P1. Line 16 : Misspelling- “30.74, SD = 5.74) in in February 2021 (T1) and may-June 2021 (T2).” 

2)P3-4: The Cronbachs α of the Perceived Negative Impact of the COVID-19 Pandemic and the Self-Reported Exposure to COVID-19 should be provided.

3)Please supplement the results of common method bias analysis in the “Statistical Analyses” section.

4) The statistic indicatorpshould be italicized, e.g. p < 0.001.

Author Response

Author Response to Reviewer 1

 We want to thank the Reviewers for the detailed evaluation of our manuscript and constructive comments. We do appreciate the valuable contribution and believe that it has helped to improve our manuscript. We have addressed and incorporated all of the Reviewers’ comments. Our responses are marked in italic and presented in bullet points.

Reviewer 1.

Thanks for the Editor for giving me this opportunity to review this article. I think this is an important article on Coronavirus-related PTSD. The literature review is very detailed, and the basis for raising issues is very sufficient. The methods are also adequately described. The results are clearly and briefly presented. The discussion was structured, and the conclusions are supported by the results.

  • We would like to thank the Reviewer for constructive feedback.

I only have a few minor comments.

   1)P1. Line 16 : Misspelling- “30.74, SD = 5.74) in in February 2021 (T1) and may-June 2021 (T2).” 

  • Thank you, it was improved.

   2)P3-4: The Cronbach’s α of the Perceived Negative Impact of the COVID-19 Pandemic and the Self-Reported Exposure to COVID-19 should be provided.

  • Thank you for noticing this issue. We added Cronbach’s α for PNIC-SES and PNIC-SR in two time points of measurements (lines 163-164).

  3)Please supplement the results of common method bias analysis in the “Statistical Analyses” section.

  • Thank you for this remark. We added a paragraph in the Statistical Analysis (2.4.1. Common method bias, lines 276-291) and in the study design part regarding controlling for acquiescence biases (lines 103-105).

   4) The statistic indicator“p”should be italicized, e.g. p < 0.001.

  • Yes, we fully agree with this comment.

Reviewer 2 Report

This is an ambitious study that included many variables, the fact that it is longitudinal, with the difficulties that this entails, makes this type of study especially valuable. Good extensive statistical analysis. The text should be revised for typographic errors

This study on the effects that the Covid19 pandemic has on young adult population deals with a relevant topic. Although the peaks of the pandemic seem to have been overcome in the countries evaluated in the study, part of the population continues to suffer its consequences. Meanwhile, in other countries with low vaccination rates there is still the possibility of new spikes.

The main strength of the study is that it has a sample of different nationalities that offers cross-cultural information. Although all the subjects belong to western developed countries (Israel being considered as such), the results can help to test hypotheses about what to be expected in other regions of the world. The study also has good statistical analysis.

However, the paper can be improved:

Typographic errors:

Line 10: replace preditor for predictor

Line 16: word repeated

Content:

Introduction

Justify why in a sample of people up to 40 years old, student status was chosen as a variable instead of professional status (line 88)

Method:

It is not specified if the respondents of the second measure are the same T1 subjects that volunteered to participate a second time, if so, was the study pseudonymized? (Lines 94-107)

Tables:

Reference is made in the text to a supplementary table with demographic information, (lines 121,122) however that table is not included.

Results:

(Lines 296-297) The clarity of exposure of negative correlations could be improved

Please see the attachment for details.

Author Response

Author Response to Reviewer 2

 We want to thank the Reviewers for the detailed evaluation of our manuscript and constructive comments. We do appreciate the valuable contribution and believe that it has helped to improve our manuscript. We have addressed and incorporated all of the Reviewers’ comments. Our responses are marked in italic and presented in bullet points.

Reviewer 2.

This is an ambitious study that included many variables, the fact that it is longitudinal, with the difficulties that this entails, makes this type of study especially valuable. Good extensive statistical analysis. The text should be revised for typographic errors

 This study on the effects that the Covid19 pandemic has on young adult population deals with a relevant topic. Although the peaks of the pandemic seem to have been overcome in the countries evaluated in the study, part of the population continues to suffer its consequences. Meanwhile, in other countries with low vaccination rates there is still the possibility of new spikes.

The main strength of the study is that it has a sample of different nationalities that offers cross-cultural information. Although all the subjects belong to western developed countries (Israel being considered as such), the results can help to test hypotheses about what to be expected in other regions of the world. The study also has good statistical analysis.

  • Thank you for providing a positive evaluation of our work.

However, the paper can be improved:

Typographic errors:

Line 10: replace preditor for predictor

Line 16: word repeated

  • Thank you for noticing this. The paper underwent an extensive English correction, and all typos have been corrected.

Content:

Introduction

Justify why in a sample of people up to 40 years old, student status was chosen as a variable instead of professional status (line 88)

  • Thank you for this justified question. In our previous research, regarding sociodemographic predictors measured with generalized estimating equations, the student status was strongly related to depression and suicidal thoughts, while employment status was not related to any of the mental health indices [1]. Moreover, the literature review shows that student status is a risk factor for mental health deterioration during the COVID-19 pandemic. Therefore, we decided to verify the student status.
  • Benatov, J.; Ochnik, D.; Rogowska, A.M.; Arzenšek, A.; Mars Bitenc, U. Prevalence and Sociodemographic Predictors of Mental Health in a Representative Sample of Young Adults from Germany, Israel, Poland, and Slovenia: A Longitudinal Study during the COVID-19 Pandemic.  J. Environ. Res. Public Health202219, 1334. https://doi.org/10.3390/ijerph19031334

Method:

It is not specified if the respondents of the second measure are the same T1 subjects that volunteered to participate a second time, if so, was the study pseudonymized? (Lines 94-107)

  • Thank you for this remark. We added information to clarify this issue in line 11: (“Therefore, the final sample comprised 1723 respondents who participated in both time points of measurement”). The study was conducted by ARIADNA Panel, which gave individual codes for each person, with respect for anonymity. We received no information (e.g., e-mail address) that would allow for the identification of participants.

Tables:

Reference is made in the text to a supplementary table with demographic information, (lines 121,122) however that table is not included.

  • Table S1, with extensive information regarding gender, age, place of residence, and employment status, has been included.

Results:

(Lines 296-297) The clarity of exposure of negative correlations could be improved

  • Thank you for this remark. We added more in-depth information regarding correlations with coronavirus-related PTSD at T1 and T2. (lines 340-343).